# Aurones and Flavonols from *Coreopsis lanceolata* L. Flowers and Their Anti-Oxidant, Pro-Inflammatory Inhibition Effects, and Recovery Effects on Alloxan-Induced Pancreatic Islets in Zebrafish

**DOI:** 10.3390/molecules26206098

**Published:** 2021-10-09

**Authors:** Hyoung-Geun Kim, Youn Hee Nam, Young Sung Jung, Seon Min Oh, Trong Nguyen Nguyen, Min-Ho Lee, Dae-Ok Kim, Tong Ho Kang, Dae Young Lee, Nam-In Baek

**Affiliations:** 1Graduate School of Biotechnology and Department of Oriental Medicinal Biotechnology, Kyung Hee University, Yongin 17104, Korea; zwang05@khu.ac.kr (H.-G.K.); 01030084217@hanmail.net (Y.H.N.); seonmin88@khu.ac.kr (S.M.O.); ntnguyen@khu.ac.kr (T.N.N.); panjae@khu.ac.kr (T.H.K.); 2Department of Food Science and Biotechnology, Kyung Hee University, Yongin 17104, Korea; chembio@khu.ac.kr (Y.S.J.); DOKIM05@khu.ac.kr (D.-O.K.); 3Department of Herbal Crop Research, National Institute of Horticultural and Herbal Science, RDA, Eumseong 27709, Korea; 4Department of Food Technology and Services, Eulji University, Seongnam 13135, Korea; minho@eulji.ac.kr

**Keywords:** antioxidant capacity, anti-inflammation, *Coreopsis lanceolata*, flavonoid, pancreatic islets, RAW 264.7, zebrafish

## Abstract

(1) Background: Many flavonoids have been reported to exhibit pharmacological activity; a preparatory study confirmed that *Coreopsis lanceolata* flowers (CLFs) contained high flavonoid structure content; (2) Methods: CLFs were extracted in aqueous methanol (MeOH:H_2_O = 4:1) and fractionated into acetic ester (EtOAc), normal butanol (*n*-BuOH), and H_2_O fractions. Repeated column chromatographies for two fractions led to the isolation of two aurones and two flavonols; (3) Results: Four flavonoids were identified based on a variety of spectroscopic data analyses to be leptosidin (**1**), leptosin (**2**), isoquercetin (**3**), and astragalin (**4**), respectively. This is the first report for isolation of **2**–**4** from CLFs. High-performance liquid chromatography (HPLC) analysis determined the content levels of compounds **1**–**4** in the MeOH extract to be 2.8 ± 0.3 mg/g (**1**), 17.9 ± 0.9 mg/g (**2**), 3.0 ± 0.2 mg/g (**3**), and 10.9 ± 0.9 mg/g (**4**), respectively. All isolated compounds showed radical scavenging activities and recovery activities in Caco-2, RAW264.7, PC-12, and HepG2 cells against reactive oxygen species. MeOH extract, EtOAc fraction, and **1**–**3** suppressed NO formation in LPS-stimulated RAW 264.7 cells and decreased iNOS and COX-2 expression. Furthermore, all compounds recovered the pancreatic islets damaged by alloxan treatment in zebrafish; (4) Conclusions: The outcome proposes **1**–**4** to serve as components of CLFs in standardizing anti-oxidant, pro-inflammatory inhibition, and potential anti-diabetic agents.

## 1. Introduction

Humans living in the modern era enjoy many advantages such as economical leisure, improvement of living standard, and growth of medicine. However, at the same time, these benefits have led to the birth of an aging society, increased stress, and caused many chronic diseases. In particular, human beings suffer from chronic metabolic diseases and so-called adult diseases, and diabetes is the most representative of these diseases. Diabetes mellitus is a metabolic disease caused by hyperglycemia, which affects more than 400 million people worldwide [1,2]. Diabetes is a disease with a high concentration of glucose in the blood due to a lack of insulin secretion or poor functioning [3]. These types of diabetes include type 1 and type 2 diabetes [4]. Type 1 diabetes results from the β-cell destruction leading to deficiency of insulin, and then high glucose in blood cannot get into cells, resulting in various symptom of diabetes [5]. However, type 2 diabetes is caused by insulin resistance and defects in insulin secretion. Then, blood glucose and insulin concentrations rise abnormally and persistent hyperglycemia causes complications that result in retinopathy and nephropathy [6]. Currently, there is no development of a method for the treatment of diabetes, and it is best to keep it at the normal blood glucose level through the hypoglycemic agent [6,7]. Acarbose, voglibose, and other synthetic hypoglycemic agents are commonly used [8], but they have a high risk of hypoglycemia and may have side effects [9]. Therefore, in order to overcome the side effects of synthetic hypoglycemic agents, researches on diabetes prevention and adjuvant therapy using natural materials have been actively carried out.

Many studies insisted that diabetic patients were exposed to oxidative stress due to abnormal activation of the superoxide dismutase (SOD) enzyme and reduction in the antioxidant, such as reduced glutathione [10]. Such increased oxidative stress causes dysfunction of pancreatic β-cells and results in insulin secretory disorder.

Recent studies proved that reducing the inflammatory response by macrophages reduces insulin resistance and treats type 2 diabetes [11]. Macrophages are typical immune cells that cause an inflammatory reaction. They infiltrate tissues, such as liver and muscle, to secrete cytokines and induce inflammatory responses. Therefore, materials inhibiting nitric oxide (NO) production by repressing inducible nitric oxide synthase (iNOS) and cyclooxygenase-2 (COX-2) enzymes can be used as adjuvant anti-type 2 diabetic agents. 

The zebrafish model has been widely used in the search for anti-diabetic drugs because of its physiological and genetic similarities to mammalian systems [12,13]. In addition, it is easy conserve in laboratories, multitudinous progeny, pellucid embryos, and amenability to genetic and chemical screens [14,15]. In order to study diabetes, zebrafish can cause variations in the size of pancreatic islets (PI) and in glucose absorption by treating alloxan (AX) to damage the PI.

Methanolic extract and their fractions of *Coreopsis lanceolata* flower (CLFs) showed DPPH and ABTS radical scavenging capacities, repression in nitrile oxide (NO) production in lipopolysaccharide (LPS)-induced RAW 264.7, reduction in iNOS and COX-2 expression in RAW264.7 cells, and recovery effect on AX-induced PI in zebrafish. Thus, this paper studied to clarify the antioxidant, anti-inflammatory, and anti-diabetes constituents in *C. lanceolata* flowers was carried out.

*C. lanceolata* (Asteraceae) is a herbaceous perennial plant originated in America, South Africa, and Eastern Asia continents, and is previously reported to have antioxidant [16], antibacterial, antiallergic [17], antileukemia [18], and nematicidal [19] effects. Notwithstanding several reported pharmacological activities of *C. lanceolata*, there are only a few studies to isolate flavonoids in *C. lanceolata* flowers [20]. 

Accordingly, we started exploring the principal components in *C. lanceolata* flowers. Finally, two aurones (**1** and **2**) and two flavonols (**3** and **4**) were isolated and identified. This is the first report for isolation of **2**–**4** from *C. lanceolata* flowers. All of the solvent fractions and flavonoids were measured for antioxidant and anti-inflammatory effects using enteric epithelial cells (Caco-2), macrophage cells (RAW264.7), neuron cells (PC-12), and hepatic cells (HepG2) from various angles (epithelium-blood-organ). Furthermore, we not only evaluated the inhibition effects on LPS-induced NO production in RAW264.7 macrophage by repressing iNOS and COX-2 but also for protective activity against alloxan-induced damage to the pancreatic islets of zebrafish larvae. The manuscript details the strategy for isolation and identification of the compounds but also their anti-oxidant, pro-inflammatory inhibition effects, and potential anti-diabetic strength. This suggested the potentiality of *C. lanceolata* flowers as nutraceuticals for anti-diabetic agents.

## 2. Results and Discussion

### 2.1. Structure Elucidation

*C. lanceolata* flowers were extracted in aqueous methanol (MeOH:H_2_O = 4:1), and the condensed extracts were segmentationed into EtOAc, *n*-BuOH, and H_2_O fractions, successively. Among the fractions, the EtOAc and *n*-BuOH fractions were used for isolation of metabolites through column chromatography to yield two aurones (**1** and **2)** and two flavonols (**3** and **4)**. All the isolated compounds **1**–**4** showed UV absorption (254 nm and 360 nm) and exhibited red (**1** and **2**) and yellow (**3** and **4**) color on the TLC plate upon spraying with 10% H_2_SO_4_ and heating, suggesting them to be flavonoids. Comparing the 1D-, 2D-NMR and FAB/MS data with reported values allowed us to identify two known flavonols, isoquercetin (**3**) [21] and astragalin (**4**) [22] (Figure 1).

Compound **1**, red amorphous powder, was given a molecular weight (MW) of 300, based on the detected molecular ion peak (MIP) [M + H]^+^
*m*/*z* 301 in the positive FAB/MS spectrum (FAB^+^). FT-IR data exhibited the absorption (cm^-1^) of hydroxy (3366), conjugated ketone (1661), and aromatic double bond (1604). Based on gHSQC spectra, the mentioned ^1^H-NMR (PMR) and ^13^C-NMR (CMR) data indicated that **1** was an aurone. PMR displayed the signals of three olefinic methines [δ_H_ 6.84 (coupling pattern, coupling constant in Hz), d, 8.2, H-5′; δ_H_ 7.26, dd, 8.2, 1.6, H-6′; δ_H_ 7.46, d, 1.6, H-2′) derived from a 1,2,4-trisubstituted benzene moiety; two olefinic methines (δ_H_ 6.73, d, 8.4, H-6; δ_H_ 7.33, d, 8.4, H-5), owing to a 1,2,3,4-tetrasubstituted benzene moiety; one olefinic methine (δ_H_ 6.70, s, H-2); and one methoxy (δ_H_ 4.12, s, 8-OCH_3_) were observed. The above mentioned PMR indicated that **1** was an aurone with a methoxy group. CMR showed 16 carbon signals, confirming **1** to be composed of an aurone and one methoxy group (δ_C_ 61.7, 8-OCH_3_) including one conjugated ketone (δ_C_ 184.6), six oxygen substituted olefinic quaternaries (δ_C_ 133.94, 146.89, 147.62, 149.71, 159.62, 159.92), two olefinic quaternaries (δ_C_ 114.70, 125.63), and six olefinic methines (δ_C_ 114.70, 115.23, 116.78, 119.26, 120.81, 126.42). CMR indicated **1** to be an aurone with three hydroxy and one methoxy groups. gHMBC correlation of the methoxy proton (δ_H_ 4.12) with the oxygen-substituted olefinic quaternary carbon (δ_C_ 133.94, C-8) which exhibited disclosing the methoxy to be at C-8. Consequently, **1** was identified to be 8-methoxymaritimetin (leptosidin), which was confirmed by comparing literature data [23] (Figure 1). Leptosidin was first isolated from *Flemengia strobilifera* in 1975 [24], and was reported to exhibit antioxidant activity [23].

Compound **2**, red amorphous powder, was decided for MW to be 462, based on MIP [M + H]^+^
*m/z* 463 in the FAB^+^. FT−IR data showed the absorption (cm^−1^) of hydroxy (3360), conjugated ketone (1659), and an aromatic double bond (1617). The 1D-NMR spectra of **2** were resembled nearly to those of **1** with the preclusion of one additional hexose, including a hemiacetal (δ_H_ 5.10, d, 7.0, H-1″; δ_C_ 102.54 ppm); four oxygen substituted methines [(δ_H_ 3.43, overlap, H-4″; δ_C_ 71.23 ppm), (δ_H_ 3.49, overlap, H-5″; δ_C_ 77.91 ppm), (δ_H_ 3.50, overlap, H-3″; δ_C_ 78.45 ppm), (δ_H_ 3.55, overlap, H-2″; δ_C_ 74.79 ppm)]; and one oxygen substituted methylene (δ_H_ 3.71, dd, 12.0, 5.2, H-6″b; 3.91, dd, 12.0, 1.2, H-6″a; δ_C_ 62.42 ppm) signal. The *J* value (7.0 Hz) of the anomeric proton proved that the anomeric hydroxy have a β-configuration. The gHMBC correlation of the anomer proton (δ_H_ 5.10) with the oxygenated olefinic quaternary carbon (δ_C_ 158.11, C-7) exhibited disclosing the presence of β-d-glucopyranose at C-7. Accordingly, **2** was identified as a leptosidin 7-*O*-β-d-glucopyranoside (leptosin), which was first isolated from *Coreopsis grandiflora* in 1943 [25] (Figure 1), and has not been reported so far any activity.

### 2.2. Quantitative Analysis of Compounds ***1**–**4*** Using HPLC

High-performance liquid chromatography (HPLC) analysis was conducted to determine the content of compounds **1**–**4** in the MeOH extract from *C. lanceolata* flowers. Gradient elution using H_2_O (+0.1% formic acid) and acetonitrile (ACN) was performed, using a XBridge C18 5 *μ*m LC column. Peaks were detected using photo diode arrays at 360 nm. Peaks appeared at 16.80, 17.53, 19.48, and 35.72 min, and were identified as those of isoquercetin (**3**), leptosin (**2**), astragalin (**4**), and leptosidin (**1**), respectively (Figure 2). Extrapolation from the regression curve of compounds determined the content of compounds **1**–**4** in the extract from *C. lanceolata* flowers to be 2.8 ± 0.3 mg/g (**1**), 18.0 ± 0.9 mg/g (**2**), 3.0 ± 0.2 mg/g (**3**), and 10.9 ± 0.9 mg/g (**4**), respectively (Table 1). This analytic method was dependable for the *r*^2^ values of regression curves for all compounds were higher than 0.999. Quantitative analysis was replicated three times.

### 2.3. Radical Scavenging Assays for Extract, Solvent Fractions, and Flavonoids ***1**–**4*** Using DPPH and ABTS

The antioxidant capacities of extract (CLF), fractions (CLFE, CLFB, and CLFW), and compounds **1**–**4** of *C. lanceolata* flowers by the DPPH and ABTS assays are shown in Table 2. CLFE showed the highest scavenging capacities in DPPH and ABTS assay than other fractions. It was suggested that the CLFE mainly contained high amount of antioxidant flavonoids, such as leptosidin. Compounds **1**–**4** showed high radical scavenging capacities in order **1** > **3** > **2** > **4** in DPPH and ABTS radicals. A previous report showed that the radical scavenging capacities measured by DPPH and ABTS radicals depend on the numeric of hydroxy (OH) groups [26]. Leptosidin (**1**) showed the highest antioxidant capacity because of 3′, 4′-disubstitution of OH groups on the B-ring, forming a pyrocatechol structure, which is widely known to be the key structure with radical scavenging capacity. Compound **1** has one less sugar than others (**2**–**4**). The steric hindrance caused by sugars was reduced in compound **1**. The aglycone of compound **3**, quercetin, also has the OH group at 3′ and 4′ on the B-ring, forming a pyrocatechol structure. As shown in Table 2, the leptosidin glucoside, compound **2,** and the quercetin glycoside, compound **3**, showed two times higher activity than the kaempferol glycoside, compound **4**.

### 2.4. Inhibition Effects of Compounds ***1**–**4*** on Intracellular Oxidative Stress in PC-12, HepG2, Caco-2, and RAW264.7 Cells

The reduction products of oxygen, i.e., reactive oxygen species (ROS), are produced by metabolic processes or external factors in normal cells in the body. Most of them have an unstable state that reaches a stable state by losing or gaining electrons. These properties are known to cause damage, various diseases, inflammation, and aging by causing oxidative stress (OS) in DNA and cell membranes *in vivo* [27,28]. For measuring ROS in cells, 2′,7′-dichlorofluorescein (DCFH) diacetate is used as a typical substance. It can freely cross the cell membrane. When the acetate group is removed by esterase, it is deacetylated with non-fluorescent DCFH. Deacetylated DCFH is oxidized by ROS and, as a result, becomes a strong fluorescent DCF. Compounds **1**–**4** from CLF inhibited intracellular ROS in colon epithelial (Caco-2), macrophage (RAW 264.7), and neuronal (PC-12) cells. ROS levels in all cell lines were raised by OS (200 μM H_2_O_2_), unlike their levels in control cells (Caco-2: 265.8%, RAW264.7: 188.2%, PC-12: 137.0%, and HepG2: 138.1%). After treating the cells with 10-μM isolated compounds, the results showed that all compounds significantly lowered stress induced by ROS. In particular, the decrease rates diversified by the cell line (Figure 3). In Caco-2 colon epithelial cells, aurones **1** and **2** significantly lowered oxidative stresses, compounds **1**–**3** were effectual in RAW264.7 macrophages, and all flavonoids lowered oxidative stresses in PC-12 neuronal cells. These diverse rates also showed unambiguous structure–activity relationships. In the Caco-2 cells, aurones **1** and **2** were most effectual, indicating that the two pyrocatechol structure in the A- and B-ring decreased oxidative stresses. Aurone **2**, which has an additional glucose compared with aurone **1**, showed a little higher capacity than aurone **1**. Additionally, flavonol **3**, which has an pyrocatechol structure in the B-ring compared with flavonol **4**, showed a little higher capacity than flavonol **4**. In the RAW264.7 cells, aurones **1, 2,** and flavonol **3**, i.e., more catechol structures in the B-ring, correlated with higher recovery effects. In PC-12 and HepG2 cells, all compounds recuperated intracellular ROS to the control level, and no differences caused by disparateness in the structure were observed. Structure divergence of flavonoids have dissimilar absorption and conveyance rates in cells [29,30]. Thus, flavonoids with different substituents are thought to have different levels of access and absorption in different types of cells. In addition, the flavonoids have different antioxidant capacities depending on the position or numeric of their hydroxyl (-OH) and various other structural characteristics, such as double bond, methylation (-OCH_3_), and number of saccharide [31,32]. In consequence, even though the aglycone compounds were the strongest radical scavenging effect, aurone **2** had a higher ability to reduce ROS in cells than **1**. Thus, the ability to decrease the ROS in a particular cell depends on a combination of absorption, penetrability, and molecular antioxidant capacity [26].

### 2.5. Inhibition Effects of Extract, Solvent Fractions, and Flavonoids ***1**–**4*** on NO formation in RAW 264.7 Macrophages

#### 2.5.1. Cytotoxicity for Extract, Fractions, and Flavonoids **1**–**4** in RAW 264.7 Macrophages

To evaluate cytotoxicity for extract, fractions, as well as isolated compounds, an MTT assay was conducted. The results showed that extract (CLF), EtOAc fraction (CLFE), *n*-BuOH fraction (CLFB), and H_2_O fraction (CLFW) had no cytotoxicity up to 100 μg/mL in RAW 264.7 cells. Furthermore, compounds **1**–**4** had no cytotoxicity up to 100 μM, either (Appendix A). 

#### 2.5.2. Inhibition Effects of Extract, Solvent Fractions, and Flavonoids **1**–**4** on NO formation in RAW 264.7 Macrophages

Stuehr and Marletta reported that nitric oxide (NO) is produced by mouse macrophage in response to bacterial lipopolysaccharide (LPS) [33]. Although the role of NO prevents the host from the invader, such as bacteria, excessive NO production is able to cause chronic inflammation [34]. Thus, the suppression effect of extract, solvent fractions, and compounds **1**–**4** on NO formation in RAW 264.7 induced with LPS was measured. Extract and CLFE dose-dependently showed the suppression effect on NO production (Figure 4). Except for astragalin (**4**), all isolated compounds **1**–**3** significantly inhibited NO production. A total of 100 μM of flavonoids **1**–**4** showed the NO production by 32.9 ± 1.1%, 49.5 ± 0.7%, 62.4 ± 0.3%, and 125.7 ± 1.5%, respectively, compared with LPS-treated cells (100%) (Figure 5). A distinct structure–activity relationship was observed in this case also. Aurones **1** and **2** showed higher efficacy in inhibiting NO production than flavonols **3** and **4** because of the existence of a catechol group. In addition, flavonol **3**, which has a catechol structure in the B-ring compared with flavonol **4**, showed a little higher efficacy in inhibiting NO production than flavonol **4**. The pyrocatechol group was previously reported to inhibit NO production through the inhibition of LPS signaling and direct scavenging of NO [35].

### 2.6. Effects of Ethyl Acetate Fraction (CLFE), and Flavonoids ***1**–**4*** on Levels of Tumor Necrosis Factor (TNF)-α, Interleukin (IL)-1β, and Interleukin (IL)-6 in LPS-Stimulated RAW264.7 Cells

Macrophages and monocytes are representative immune cells, and they are activated by components derived from invading bacteria or cytokines secreted by other immune cells in the body to induce an effective inflammatory response. Stimulation of lipopolysaccharide (LPS) secretes inflammation-inducing cytokines, such as tumor necrosis factor (TNF)-α, interleukin (IL)-6, and IL-1β from macrophages (including RAW 264.7 cells). Therefore, we assessed the inhibitory effects of CLFE (100 μg/mL) and flavonoids **1**–**4** (100 μM) on the LPS-induced production of TNF-*α*, IL-6, and IL-1*β*. RAW264.7 cells were incubated with CLFE and flavonoids **1**–**4** for 3 h and then stimulated with LPS for 24 h. Figure 6 showed that CLFE and flavonoids **1**–**4** significantly repressed TNF-*α*, IL-6, and IL-1*β* in RAW 264.7 cells.

### 2.7. Inhibition Effects of Extract, Solvent Fractions, and Flavonoids ***1**–**4*** on Expression of iNOS and COX-2 in RAW264.7 Cells

Once inflammation occurred by inflammatory stimuli, iNOS continuously produces high levels of NO [36]. Furthermore, NO activates COX-2, which induces the release of pro-inflammatory prostaglandins [37]. Because extract, CLFE, leptosidin (**1**), leptosin (**2**), and isoquercetin (**3**) inhibited NO production, we hypothesized that they would also reduce the production of iNOS and COX-2. As expected, they were suppressed by compounds **1**–**3**. The protein levels of iNOS and COX-2 were decreased by treatments with extract (74.9 ± 3.3% and 91.6 ± 3.5%), CLFE (45.3 ± 1.7% and 60.1 ± 3.1%), CLFB (80.8 ± 2.5% and 77.5 ± 2.3%), CLFW (74.3 ± 2.1% and 50.7 ± 2.2%), leptosidin (**1**) (55.5 ± 1.6% and 59.1 ± 3.3%), leptosin (**2**) (76.2 ± 2.6% and 89.9 ± 4.4%), isoquercetin (**3**) (77.9 ± 1.1% and 66.2 ± 3.3%), and astragalin (**4**) (120.7 ± 1.3% and 110.1 ± 4.6%), compared with LPS-treated cells, respectively (Figure 7). In particular, **1**–**3** highly inhibited the expression of iNOS and COX-2 because of having catechol groups in B-ring, and compound **1** showed the highest inhibition effects because of the absence of attached sugar [38].

### 2.8. Protective Effects of Extract, Solvent Fractions, and Flavonoids ***1**–**4*** on Pancreatic Islets (PI) in Zebrafish Treated by Alloxan

Protective activity of extract, solvent fractions, and isolated flavonoids **1**–**4** was assessed against alloxan-treated PI in zebrafish. Because of their physiological resemblances to mammals, zebrafish PI was used for model type 1 and 2 diabetes with alloxan (AX), a diabetogenic material that was reported to reduce β-cell mass in PIs [39,40,41]. To assess alloxan-treated PI, the size changes of the PIs and the fluorescence intensities of NBDG-stained PIs using fluorescence microscope were analyzed. When the zebrafish larvae were exposed to alloxan, the PI size was reduced significantly by 47.8% (*p* = 0.0003) compared to the normal group (Figure 8A). Zebrafish larvae treated with glimepiride, a positive control, revealed recovery capacity against AX-treated PI by a 99.5% (*p* = 0.0047) increase compared to AX-treated group. The PI sizes in the groups treated with extract (CLF), EtOAc (CLFE), *n*-BuOH (CLFB), and H_2_O (CLFW) fractions were increased by 70.0% (*p* = 0.0091), 68.4% (*p* = 0.0065), 69.8% (*p* = 0.005), and 73.2% (*p* = 0.0083), respectively, compared to the AX-treated group (Figure 8). All of the isolated compounds also resulted in an increase in PI size with statistical significance. Flavonoids **1**–**4** (F1–F4) increased the sizes of the damaged PIs by 98.7% (*p* = 0.0037), 89.8% (*p* = 0.0012), 78.0% (*p* = 0.0002), and 87.9% (*p* = 0.0011), respectively, compared to the AX-treated group (Figure 8A). All isolated compounds in this study increased the sizes of AX-treated PIs with high levels of significance. In particular, leptosidin (**1**) displayed almost the same recovery effects as that of the positive control, i.e., glimepiride.

### 2.9. Action of Diazoxide (DZ) on Alloxan-Induced PIs in Zebrafish

In glucose-stimulated insulin secretion, metabolism of glucose in PIs is the key step [42]. Diazoxide (DZ), a K_ATP_ channel opener, was used to examine the involvement of pancreatic β-cell K_ATP_ channel stimulation activity. The PI size in the DZ-treated normal group was significantly smaller (70.4%, *p* = 0.0052) compared to the normal group. The AX group showed no significant differences compared to the AX + DZ-treated group. PI size in the 10-μg/mL glimepiride (GLM) + AX + DZ co-treatment groups was significantly lower (50.9%, *p* = 0.002) compared to the GLM + AX-treated groups. The AX + CLFW-treated groups were not significantly different after treatment with DZ, indicating that CLFW had no act as a K_ATP_ channel opener. Furthermore, groups co-treated with **1** + AX, **2** + AX, and **4** + AX were not significantly different after treatment with DZ, indicating that compounds **1**, **2**, and **4** did not act as a K_ATP_ channel opener. Moreover, the **3** + AX group showed significantly smaller PI sizes after treatment with DZ (61.6%, *p* = 0.0358) compared to the non-DZ group (Figure 8B). These results indicate that compound **3** can stimulate insulin secretion by Ca^2+^ influx via closure of K_ATP_ channels in β-cells.

## 3. Materials and Methods

### 3.1. Plant Materials

*Coreopsis lanceolata* flowers were obtained at Kyung Hee University, Yongin, Korea in 2020 and identified by Professor Dae-Keun Kim, Woosuk University, Jeonju, Korea. A voucher specimen (KHU2020-0701) was reserved at the Laboratory of Natural Products Chemistry, Kyung Hee University, Yongin, Korea.

### 3.2. General Experimental Procedures

The materials and equipment used for the isolation and structure determination of constituents are described in a previous study [27].

### 3.3. Extraciton and Isolation

Preparation procedure of MeOH extracts (CLF) and solvent fractions (Frs), ethyl acetate (CLFE 194 g), *n*-butanol (CLFB 254 g), and H_2_O (CLFW 652 g) Frs from *C. lanceolata* flowers and the isolation procedures of compounds **1**–**4** from CLF are presented in Figure 9 and Appendix A. The physico-chemical and spectroscopic characteristics of isolated flavonoids are presented in Table 3, Table 4 and Table 5.

### 3.4. Quantitative Analysis of Flavonoids ***1**–**4*** Using HPLC

The materials, equipment, and methods used for quantitative analysis of the isolated flavonoids from CLF are described in Appendix A. The solvent elution was graded as Table 6 and Appendix A. The quantitative analysis was repeated three times.

### 3.5. Antioxidant Activities

#### 3.5.1. Free Radical Scavenging Activity

The materials, equipment, and methods used for the free radical scavenging assay of extract (CLF), fractions (CLFE, CLFB, and CLFW), and flavonoids **1**–**4** from *C. lanceolata* flowers are described in Appendix A and previous studies [27,43,44].

#### 3.5.2. Cell Culture and Cytotoxicity Assessment

The materials, equipment, and methods used for the cell culture and cytotoxicity assessment of extract (CLF), fractions (CLFE, CLFB, and CLFW), and flavonoids **1**–**4** from *C. lanceolata* flowers are described in Appendix A and previous studies [27,45].

#### 3.5.3. Measurement of Intracellular Oxidative Stress

The materials, equipment, and methods used for the measurement of intracellular oxidative stress of flavonoids **1**–**4** from *C. lanceolata* flowers are described in Appendix A and previous studies [27,46].

### 3.6. Pro-Inflammatory Inhibition Activities

#### 3.6.1. Determination of NO Production

NO produced by RAW 264.7 cells was determined using a method described in Appendix A and the literature [27,47]. 

#### 3.6.2. Assays for IL-1β, IL-6, and TNF-α

The materials, equipment, and methods used for the assays for IL-1β, IL-6, and TNF-α of ethyl acetate fraction (CLFE), as well as the flavonoids **1**–**4** from *C. lanceolata* flowers, are described in Appendix A and a previous study [47].

#### 3.6.3. Western Blot Analysis for Protein Expression

The materials, equipment, and methods used for the western blot analysis for protein expression of extract (CLF), fractions (CLFE, CLFB, and CLFW), and flavonoids **1**–**4** from *C. lanceolata* flowers are described in Appendix A and previous studies [27,47].

### 3.7. Antidiabetic Activity

#### 3.7.1. Chemicals and Animals

The chemical materials used for the antidiabetic activity are described in Appendix A and previous studies [48,49,50].

#### 3.7.2. Animals

The animal preparation used for the antidiabetic activity are also described in Appendix A and previous studies [48,49,50].

#### 3.7.3. Ethics Statement

All zebrafish experimental procedures were carried out in accordance with standard zebrafish protocols and were approved by the Animal Care and Use Committee of Kyung Hee University [KHUASP(SE)-15-10].

#### 3.7.4. Evaluation of Recovery Efficacy on Pancreatic Islet Damaged by Alloxan in Zebrafish

The materials, equipment, and methods used for evaluation of recovery efficacy of extract (CLF), fractions (CLFE, CLFB, and CLFW), and flavonoids **1**–**4** from *C. lanceolata* flowers on pancreatic islet damaged by alloxan in zebrafish are described in Appendix A and a previous study [48,49,50]. 

#### 3.7.5. Action of Diazoxide on Alloxan-Induced Diabetic Zebrafish 

The materials, equipment, and methods used for action of diazoxide on alloxan-induced diabetic zebrafish are described in Appendix A and previous studies [48,49,50].

### 3.8. Statistical Anlaysis

Results (mean ± SD, *n* = 3) were assessed using one-way analysis of variance and the Tukey–Kramer honestly significant difference (HSD) test, whereby *p* < 0.05 was considered to represent statistical significance. All statistical analyses were performed using SPSS 22.0 (SPSS Inc., Chicago, IL, USA).

## 4. Conclusions

We carried out this research to find other pharmacological active compounds from *C. lanceolata* flowers. Two aurones and two flavonol glucosides were isolated through repeated column chromatography using silica gel (SiO_2_), octadecyl SiO_2_ (ODS), and Sephadex LH-20 resins, and were identified by analysis of UV, IR, NMR, and MS data. Leptosidin (1) exhibited high DPPH and ABTS radical scavenging activity. All four flavonoids showed powerful antioxidation by reducing oxidative stress in intestinal epithelial cells (Caco-2), macrophage cells (RAW264.7), neuron cells (PC-12), and hepatic cells (HepG2). Extract, EtOAc fraction, and flavonoids **1**–**3** inhibited NO production and extract, EtOAc, *n*-BuOH, H_2_O fractions, and flavonoids **1**–**3** suppressed iNOS protein expression in RAW 264.7 cells treated with LPS. Additionally, EtOAc, *n*-BuOH, H_2_O fractions, and flavonoids **1** and **3** decreased COX-2 protein expression. Extract, EtOAc fraction, and flavonoids **1** and **3** were notably capable of inhibiting the pro-inflammatroy cytokines (TNF-α, IL-1β, and IL-6). Furthermore, H_2_O fraction and all flavonoids **1**–**4** recovered the alloxan-treated PI in zebrafish with high levels of significance. In particular, isoquercetin (**3**) stimulated insulin secretion by a Ca^2+^ influx via closure of K_ATP_ channels in pancreatic *β*-cells. The data indicate that the fractions and compounds from *C. lanceolata* flowers can activate the potential antioxidant effect, immune response by inhibiting excessive inflammation, and a recovery effect on damaged pancreatic β–cells. These results indicate that *C. lanceolata* flowers and its isolated flavonoids are used as potential anti-oxidant, pro-inflammatory inhibition, and anti-diabetic agents.

## Figures and Tables

**Figure 1 molecules-26-06098-f001:**
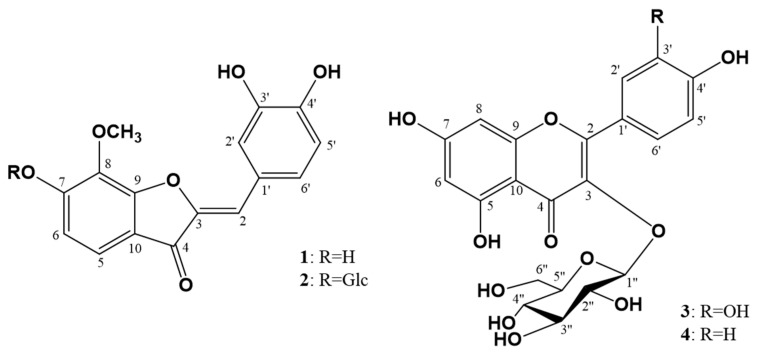
Molecular structures of compounds **1**–**4** from *Coreopsis lanceolata* flowers.

**Figure 2 molecules-26-06098-f002:**
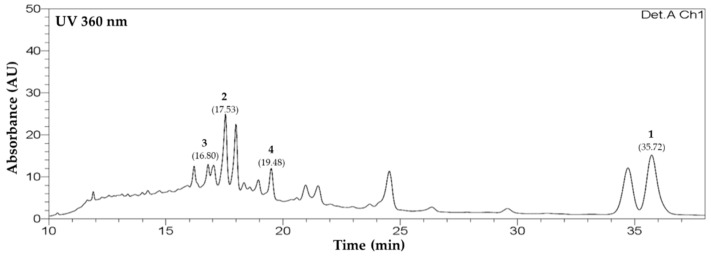
HPLC chromatograms from *Coreopsis lanceolata* L. flowers.

**Figure 3 molecules-26-06098-f003:**
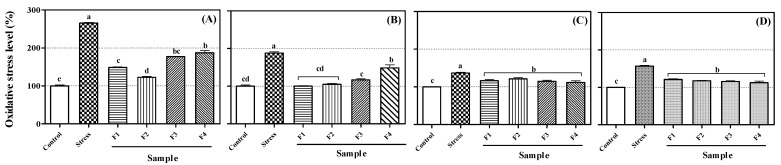
Recovery effects of flavonoids **1**–**4** (F1~F4) on intracellular oxidative stress in: (**A**) Caco-2 epithelial cells; (**B**) RAW264.7 macrophages; (**C**) PC-12 neurons; (**D**) HepG2 hepatocytes. Lowercase letters on the bars indicate significant differences according to the Tukey–Kramer HSD test (*p* < 0.05).

**Figure 4 molecules-26-06098-f004:**
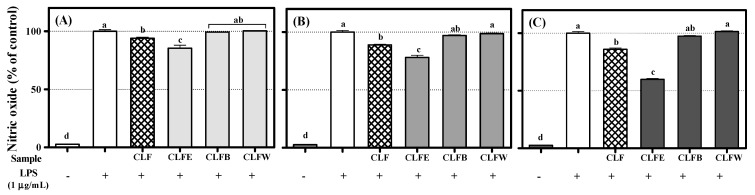
Inhibition effect of extract and solvent fractions on nitric oxide (NO) production in RAW 264.7 cells: (**A**) 25 µg/mL; (**B**) 50 μg/mL; (**C**) 100 μg/mL. Lowercase letters on the bars indicate significant differences according to the Tukey–Kramer HSD test (*p* < 0.05). CLF: *C. lanceolata* flowers 80% methanol extract, CLFE: *C. lanceolata* flowers EtOAc fraction, CLFB: *C. lanceolata* flowers *n*-BuOH fraction, CLFW: *C. lanceolata* flowers H_2_O fraction.

**Figure 5 molecules-26-06098-f005:**
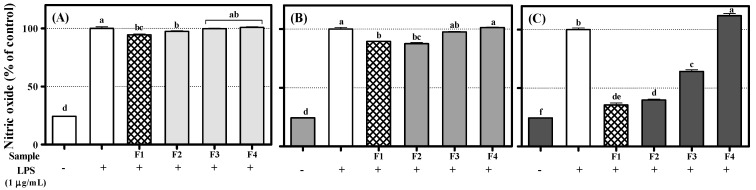
Inhibition effect of flavonoids **1**–**4** (F1~F4) on nitric oxide (NO) production in RAW 264.7 cells: (**A**) 25 μM; (**B**) 50 μM; (**C**) 100 μM. Lowercase letters on the bars indicate significant differences according to the Tukey–Kramer HSD test (*p* < 0.05).

**Figure 6 molecules-26-06098-f006:**
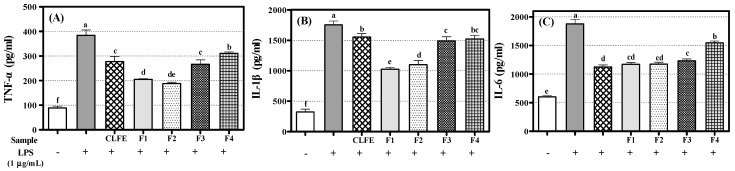
Effects of ethyl acetate fraction (CLFE), and flavonoids **1**–**4** on levels of tumor necrosis TNF-α (A), IL-1β (B), and IL-6 (C) inlipopolysaccharide (LPS)-stimulated RAW264.7 cells. Cells were pretreated for 3 h with indicated concentrations of samples and stimulated for 24 h with LPS (1 μg/mL). ELISA analysis was performed, as described in Materials and Methods. Lowercase letters on the bars indicate significant differences according to the Tukey–Kramer HSD test (*p* < 0.05).

**Figure 7 molecules-26-06098-f007:**
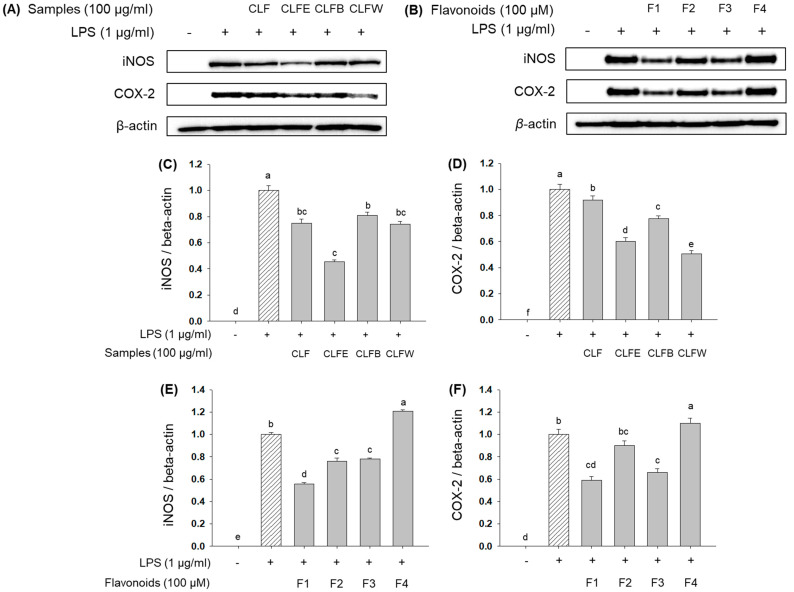
Inhibition effect of extract, solvent fractions, and flavonoids **1**–**4** (F1–F4) on the expression of iNOS and COX-2 in RAW264.7 cells: (**A**) Immunoblotting method for extract and solvent fractions; (**B**) Immunoblotting method for flavonoids **1**–**4** (F1–F4); (**C**) iNOS/β-actin ratio for extract and solvent fractions; (**D**) COX-2/β-actin ratio for extract and solvent fractions; (**E**) iNOS/β-actin ratio for flavonoids **1**–**4** (F1–F4); (**F**) COX-2/β-actin ratio for flavonoids **1**–**4** (F1~F4). Lowercase letters on the bars indicate significant differences according to the Tukey–Kramer HSD test (*p* < 0.05). CLF: *C. lanceolata* flowers 80% methanol extract, CLFE: *C. lanceolata* flowers EtOAc fraction, CLFB: *C. lanceolata* flowers *n*-BuOH fraction, CLFW: *C. lanceolata* flowers H_2_O fraction.

**Figure 8 molecules-26-06098-f008:**
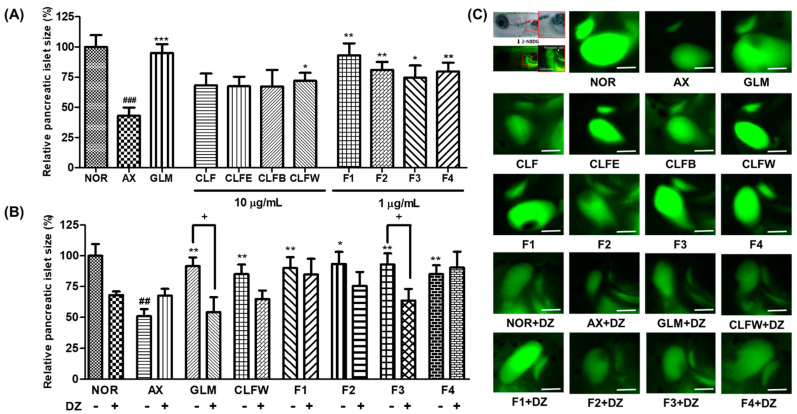
Protective effect of extract, solvent fractions, and flavonoids **1**–**4** from *Coreopsis lanceolata* flowers on damaged PIs in zebrafish treated by alloxan and action of diazoxide (DZ) for PIs damaged by alloxan (AX) in zebrafish. (**A**) Size of the PIs for protective effects; (**B**) Size of the PIs for action of DZ; (**C**) PI image: NOR: normal group, AX: alloxan group, GLM: glimepiride + AX group, CLF: extract + AX, CLFE: EtOAc fraction + AX, CLFB: *n*-BuOH fraction + AX, CLFW: H_2_O fraction + AX, F1–4: flavonoids **1**–**4** + AX. (### *p* < 0.001, ## *p* < 0.01; compared to the normal group), (* *p* < 0.05, ** *p* < 0.01, *** *p* < 0.001 compared to the alloxan-treated group). Scale bar = 100 μm.

**Figure 9 molecules-26-06098-f009:**
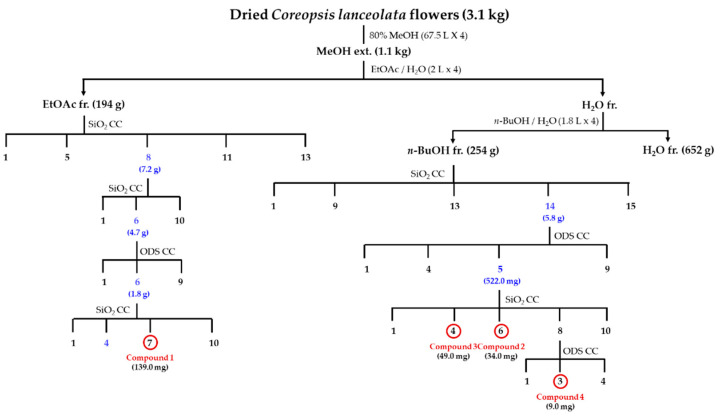
Isolation procedures of compounds **1**–**4** from *Coreopsis lanceolata* flowers. fr.: fraction; CC: column chromatography; SiO_2_: silica gel; ODS: octadecyl silica gel.

**Table 1 molecules-26-06098-t001:** Quantitative HPLC analysis for the contents of flavonoids **1**–**4** in MeOH extracts from *Coreopsis lanceolata* flowers.

Flavonoids	RT ^1^	Regression Equation	*r* ^2^	C ^2^
**1**	35.72	y = 31643x + 25702	0.9995	2.8 ± 0.3
**2**	17.53	y = 2369.8x + 680.75	0.9998	18.0 ± 0.9
**3**	16.80	y = 7887x − 61048	0.9990	3.0 ± 0.2
**4**	19.48	y = 1620.4x − 10101	0.9991	10.9 ± 0.9

^1^ Retention time (min); ^2^ Contentrations in extracts from *C. lanceolata* flowers (mg/g).

**Table 2 molecules-26-06098-t002:** DPPH and ABTS radical-scavenging activity of extract, solvent fractions, and flavonoids **1**–**4** from *Coreopsis lanceolata* flowers. CLF: *C. lanceolata* flowers 80% methanol extract, CLFE: *C. lanceolata* flowers EtOAc fraction, CLFB: *C. lanceolata* flowers n-BuOH fraction, CLFW: *C. lanceolata* flowers H_2_O fraction.

Samples	DPPH Radical (mg VCE/100g DW) ^1^	ABTS Radical (mg VCE/100g DW) ^1^
1	82.5 ± 5.3 ^a2^	1084.4 ± 0.3 ^a^
2	38.7 ± 3.2 ^b^	551.7 ± 0.8 ^b^
3	22.7 ± 1.9 ^d^	439.2 ± 0.5 ^c^
4	18.9 ± 2.0 ^e^	306.5 ± 0.2 ^d^
CLF	18.8 ± 0.8 ^e^	306.4 ± 0.3 ^d^
CLFE	32.8 ± 1.6 ^c^	549.9 ± 2.0 ^b^
CLFB	12.2 ± 2.1 ^f^	223.1 ± 0.7 ^e^
CLFW	6.5 ± 0.6 ^g^	189.4 ± 0.1 ^f^

^1^ Milligrams of vitamin C equivalent (VCE)/100g of dry weight; ^2^ Data are presented as the mean ± standard deviation (*n* = 3); Means with different superscripts in the same column differ significantly different by Tukey-Kramer’s HSD test (*p* < 0.05).

**Table 3 molecules-26-06098-t003:** Physico-chemical and spectroscopic characteristics of flavonoids from *Coreopsis lanceolata* flowers.

Comp.^1^	Crystals Characteristics	m.p. (°C)	[α]_D_ ^2^	FAB/MS ^3^	IR ^4^
**1**	Red amorphous powder	252−254	-	301 [M + H]^+^	3366, 1661, 1604
**2**	Red amorphous powder	229−231	−62.3°	463 [M + H]^+^	3360, 1659, 1617
**3**	Yellow amorphous powder	230−233	−66.2°	465 [M + H]^+^	3366, 1660, 1607, 1501
**4**	Yellow amorphous powder	218−220	−69.9°	449 [M + H]^+^	3364, 1656, 1607, 1506

^1^ Compound; ^2^
*c* = 0.10, CH_3_OH; ^3^ positive, *m/z*; ^4^ CaF_2_, *v*, cm^−1^.

**Table 4 molecules-26-06098-t004:** ^1^H NMR data for compounds **1**–**4**. δ_H_ in ppm, coupling pattern, *J* in Hz.

No. of H	1 ^a^	2 ^b^	3 ^c^	4 ^c^
2	6.70, s	6.70, s	-	-
5	7.33, d, 9.0	7.41, d, 8.4	-	-
6	6.72, d, 9.0	7.09, d, 8.4	6.24, br. s	6.69, br. s
8	-	-	6.32, br. s	6.71, br. s
2′	7.46, d, 1.2	7.45, d, 1.2	7.70, br. s	8.45, d, 8.8
3′	-	-	-	7.19, d, 8.8
5′	6.83, d, 8.4	7.10, d, 8.0	6.88, d, 8.0	7.19, d, 8.8
6′	7.26, dd, 8.4, 1.2	7.26, dd, 8.0, 1.2	7.57, br. d, 8.0	8.45, d, 8.8
8-OCH_3_	4.18, s	4.12, s	-	-
1″	-	5.10, d, 7.0	5.18, d, 7.0	6.21, d, 7.0
2″	-	3.57, dd, 7.0, 7.0	3.55, dd, 7.0, 7.0	4.20 ^#^
3″	-	3.50 ^#^	3.49, dd, 7.0, 7.0	4.19 ^#^
4″	-	3.45, dd, 7.0, 7.0	3.43 ^#^	4.01, dd, 7.0, 7.0
5″	-	3.49 ^#^	3.47 ^#^	4.18 ^#^
6″	-	3.92, dd, 12.0, 1.2 3.75, dd, 12.0, 5.2	3.98, br. d, 12.0 3.84, dd, 12.0, 5.6	4.35, br. d, 12.0 4.21, dd, 12.0, 5.6

^a^ CD_3_OD, 600 MHz; ^b^ CD_3_OD, 400 MHz; ^c^ pyridine-*d*_5_, 400 MHz; ^#^ overlapped.

**Table 5 molecules-26-06098-t005:** ^13^C NMR data for compounds **1**–**4**. δ_C_ in ppm.

No. of C	1 ^a^	2 ^b^	3 ^c^	4 ^c^
2	115.23	115.81	158.91	161.25
3	147.62	147.09	135.66	135.64
4	184.64	184.73	179.32	181.33
5	120.81	120.11	160.17	157.21
6	114.70	113.78	96.75	94.16
7	159.62	158.11	165.98	166.01
8	133.94	136.17	98.52	99.46
9	159.92	159.42	159.95	165.52
10	114.87	115.55	105.66	103.71
8-OCH_3_	61.72	61.88	-	-
1′	125.63	125.82	123.23	121.62
2′	119.26	118.97	115.99	131.44
3′	146.89	146.92	145.82	115.63
4′	149.71	149.87	150.00	164.17
5′	116.78	116.61	117.60	115.59
6′	126.42	126.75	123.21	131.41
1″	-	102.54	105.64	106.12
2″	-	74.79	75.13	75.69
3″	-	78.45	78.35	78.56
4″	-	71.23	71.11	71.31
5″	-	77.91	78.00	78.01
6″	-	62.42	62.37	62.36

^a^ CD_3_OD, 150 MHz; ^b^ CD_3_OD, 100 MHz; ^c^ pyridine-*d*_5_, 100 MHz.

**Table 6 molecules-26-06098-t006:** Solvent gradient for quantitative analysis of isolated flavonoids from *Coreopsis lanceolata* flowers using HPLC.

Time (min)	Flow rate (mL/min)	Solvent A (%)	Solvent B (%)
**0**	1	95	5
**3**	1	95	5
**6**	1	85	15
**12**	1	80	20
**35**	1	80	20
**37**	1	10	90
**40**	1	95	5
**45**	1	95	5

## Data Availability

Data is contained within the article.

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
