# Peer review of "Aurones and Flavonols from Coreopsis lanceolata L. Flowers and Their Anti-Oxidant, Pro-Inflammatory Inhibition Effects, and Recovery Effects on Alloxan-Induced Pancreatic Islets in Zebrafish"

_molecules, 2021, doi:10.3390/molecules26206098_

Round 1
Reviewer 1 Report
In the manuscript entitled “Aurones and flavonols from Coreopsis lanceolata L. flowers and 2 their anti-inflammatory and anti-diabetic effects”, the authors evaluate the effect of four isolated flavonoids, leptosidin, leptosin, isoquercetin, and astragalin. Based on several experiments carried out on different cell types, enteric epithelial cells (Caco-2), macrophage cells (RAW264.7), and neuron cells (PC-12), the authors suggest that these compounds improve ROS-induced oxidative stress, reduce nitric oxide formation, and inhibit iNOS and COX-2 expression (although astragalin seems to have no effect). In addition, all compounds recovered the pancreatic islets damaged by alloxan treatment in zebrafish. The authors indicate that all compounds could be used as anti-oxidant, anti-inflammatory, and anti-diabetic agents (page 13, line 414).
- The authors use Caco-2, RAW264.7, and PC-12 cells. They should use other cell lines which are targets of insulin signaling, such as hepatic and muscle cells or adipocytes.
- The authors conclude that these compounds have some anti-inflammatory effects. To this end, the authors should measure the expression of different cytokines, such as IL-1, IL-6, TNFalpha, IFN-gamma or TGFbeta. In addition, they should measure the levels of anti-inflammatory cytokines such as IL-10. The authors have to perform all of these experiments for their conclusions.
- In fig. 5, differences on nitric oxide production seem sometimes minimal, almost imperceptible. Are the authors confident with these results?
- The authors indicate that all compounds could be used as anti-diabetic agents, without measuring insulin signaling in insulin responsive cells (hepatic, muscle cells or adipocytes). Is insulin receptor andr IRSs tyrosine phosphorylation induced? What happens to Akt/PKB or ERK1/2 phosphorylation? Are glucose uptake or glycogen synthesis improved? The authors have to perform all of these experiments for their conclusions.
- In figure 6, the authors should measure also iNOS phosphorylation.
- Please proofread and fix some typing error throughout the manuscript. Please, focused also on standard English revision.
Author Response
#Reviewer 1
Comments and Suggestions for Authors
In the manuscript entitled “Aurones and flavonols from Coreopsis lanceolata L. flowers and 2 their anti-inflammatory and anti-diabetic effects”, the authors evaluate the effect of four isolated flavonoids, leptosidin, leptosin, isoquercetin, and astragalin. Based on several experiments carried out on different cell types, enteric epithelial cells (Caco-2), macrophage cells (RAW264.7), and neuron cells (PC-12), the authors suggest that these compounds improve ROS-induced oxidative stress, reduce nitric oxide formation, and inhibit iNOS and COX-2 expression (although astragalin seems to have no effect). In addition, all compounds recovered the pancreatic islets damaged by alloxan treatment in zebrafish. The authors indicate that all compounds could be used as anti-oxidant, anti-inflammatory, and anti-diabetic agents (page 13, line 414).
# The authors use Caco-2, RAW264.7, and PC-12 cells. They should use other cell lines which are targets of insulin signaling, such as hepatic and muscle cells or adipocytes.
- As you pointed, The inhibition effect of samples on intercellular oxidative stress in hepatic cell (HepG2) was added.
# The authors conclude that these compounds have some anti-inflammatory effects. To this end, the authors should measure the expression of different cytokines, such as IL-1, IL-6, TNFalpha, IFN-gamma or TGFbeta. In addition, they should measure the levels of anti-inflammatory cytokines such as IL-10. The authors have to perform all of these experiments for their conclusions.
- Thank you for your meaningful comment. We also agree with you. We also wanted to do the experiment using all cytokines as in your comments, but the amount of each compound was not enough for further testing. Therefore, we evaluated inhibition effects ethyl acetate fraction (CLFE), and flavonoids 1–4 on levels of TNF-α, IL-1β, and IL-6. And the results added in manuscript (sub-chapter 2.6) and the material and methods added in manuscript (sub-chapter 3.6.2). We sincerely hope that this answer was appropriate to your comments and sincerely hope that you will be fully understand our situation.
# In fig. 5, differences on nitric oxide production seem sometimes minimal, almost imperceptible. Are the authors confident with these results?
- Your indication is absolutely right. The concentration of treated compounds for antioxidant activity evaluation is 10 μM, but that for inhibition test on NO production is 25 to 100 μM. Both the experiments were carried out at the same time. 25-100 μM of the isolated compounds were used for activity test relating to inflammation. Usually, most well known positive controls to inhibit NO production are reported to be ca 10 μM of IC50 values. And some naturally occurred compounds to be acknowledged inhibiting NO production were reported to be ca 20 to 60 μM of IC50 values. Therefore, successive study is necessary using lower concentration of the compounds, solvent extracts, or fractions to develop antioxidant or anti-inflammation materials. So, I replaced the anti-inflammatory activity by a pro-inflammatory inhibition activities (sub-chapter 3.6). I hope this paper is thought to be the preliminary for activity study.
# The authors indicate that all compounds could be used as anti-diabetic agents, without measuring insulin signaling in insulin responsive cells (hepatic, muscle cells or adipocytes). Is insulin receptor andr IRSs tyrosine phosphorylation induced? What happens to Akt/PKB or ERK1/2 phosphorylation? Are glucose uptake or glycogen synthesis improved? The authors have to perform all of these experiments for their conclusions.
- Thank you for this comment. In this study, we focused on the pro-inflammatory inhibition and anti-diabetic effects of components of CLFs and conducted the experiment to find out the anti-diabetic effect by evaluating the recovery effect on damaged pancreatic islets. In the end, we confirmed the components of CLFs has anti-diabetic effect, Therefore, our project is getting ready to submit the follow-up article by adding the data of the anti-diabetes mechanism including insulin receptor and glucose uptake in detail.
# In figure 6, the authors should measure also iNOS phosphorylation.
- Thank you for your meaningful comment. As in the previous answer, it is difficult to doing the further study because the amount of the compound is insufficient to test. However, we believe that it might be able to be sufficient meaningful on the present results of suppression of the production of inflammatory cytokines and the suppression of iNOS and COX-2 protein expression. Nevertheless, the focus of this manuscript is to report for the first time that two aurones and flavonols were isolated from C. lanceolata flowers, and these flavonoids siginificantly showed anti-oxidant, pro-inflammatory inhibition, and anti-diabetic actions. Therefore, we think that the present results are also enough meaningful for basic research of compounds isolated from C. lanceolata flowers for the developments of anti-oxidant, pro-inflammatory inhibition, and anti-diabetic reagents. We sincerely hope that this answer was appropriate to your comments and sincerely hope that you will be fully understand our situation.
# Please proofread and fix some typing error throughout the manuscript. Please, focused also on standard English revision.
- This manuscript was previously checked by professional company dealing with English before submission. Please refer to the complementary file. But because it was suggested for revision of English, we asked a native speaker to revise English of the revised manuscript. And as you pointed, typing errors were all corrected. Thank you for your delicate revision.

Reviewer 2 Report
1/ In the Introduction you write: [Lines 43-45:] In diabetic patients, postprandial blood sugar and insulin concentrations rise abnormally and persistent hyperglycemia causes complications that result in retinopathy and nephropathy. Is insuline concentration rise true for the type 1 diabetes?
2/ Lines 54-55: Such increased oxidative stress causes dysfunction of β-cells and results in insulin secretory disorder.
Please, insert "pancreatic" before "β-cells".
3/ NO is "nitric oxide" not "nitrile oxide" [Lines 59-60] or "nitrite..." [Figures 4 and 5].
4/ Table 2: In the Table caption there is "CLF", in the table itself there is "extraxt". Please, unify over the whole text.
5/ Please, separate results of the cytotoxicity experiments from NO production [sub-chapter 2.5.].
6/ Lines 198, 199, 202: Please, put "oxidative stress" instead of "ROS stresses".
7/ Analysis of methods listed in sub-chapters 3.5, 3.6, 3.7 needs following 2 additional articles [references 24 and 40]. Please, put a general principle of each method.
8/ Figure 6 refers to the immunoblotting results, not to the AX treatment [Lines 289 and 292]
9/ In the Figure 7C caption please, put the name of the fluorescent dye [2-NBDG]. Which statistical test was applied to assess the results? Where is the scale bar? The picture in the Figure 7C [top left corner] resembles very much the picture in the Figure 1B in the Reference 40. Can you provide an original material or mention the source?
Author Response
#Reviewer 2
Comments and Suggestions for Authors
1/ In the Introduction you write: [Lines 43-45:] In diabetic patients, postprandial blood sugar and insulin concentrations rise abnormally and persistent hyperglycemia causes complications that result in retinopathy and nephropathy. Is insuline concentration rise true for the type 1 diabetes?
- Thank you for this comment. I have added description in introduction.
Diabetes is a disease with a high concentration of glucose in the blood due to a lack of insulin secretion or poor functioning [3]. These types of diabetes include type 1 and type 2 diabetes [4]. Type 1 diabetes results from the B-cell destruction leading to deficiency of insulin and then high glucose in blood can’t get into cells, resulting in various symptom of diabetes [5]. However, Type 2 diabetes is caused by defects in insulin resistance and progressive insulin secretion. Then, blood glucose and insulin concentrations rise abnormally and persistent hyperglycemia causes complications that result in retinopathy and nephropathy [6].
2/ Lines 54-55: Such increased oxidative stress causes dysfunction of β-cells and results in insulin secretory disorder. Please, insert "pancreatic" before "β-cells".
- As you pointed, we have revised the manuscript to insert “pancreatic” in Line 54-55.
3/ NO is "nitric oxide" not "nitrile oxide" [Lines 59-60] or "nitrite..." [Figures 4 and 5].
- We are sorry for confusion caused by our mistake. We all revised the mistyping “nitrile, nitrite” to “nitric”.
4/ Table 2: In the Table caption there is "CLF", in the table itself there is "extraxt". Please, unify over the whole text.
- As you pointed, we have revised the manuscript to unify “CLF” in table caption and table.
5/ Please, separate results of the cytotoxicity experiments from NO production [sub-chapter 2.5.].
- As you pointed, we have revised the manuscript to separate cytotoxicity experiments (sub-chapter 2.5.1.) and NO production (sub-chapter 2.5.2.).
6/ Lines 198, 199, 202: Please, put "oxidative stress" instead of "ROS stresses".
- As you pointed, we have revised the manuscript to insert “oxidative stress” in Line 198, 199, 202.
7/ Analysis of methods listed in sub-chapters 3.5, 3.6, 3.7 needs following 2 additional articles [references 24 and 40]. Please, put a general principle of each method.
- As you pointed, we have revised the manuscript to insert reference of general principles in 3.5, 3.6, and 3.7. Also, The general principle of each methods were describe in supplementary material S8.
8/ Figure 6 refers to the immunoblotting results, not to the AX treatment [Lines 289 and 292]
- We are sorry for confusion caused by our mistake. Figure 8 in revised manuscript refers to the AX treatment. Therefore, we have revised the manuscript.
9/ In the Figure 7C caption please, put the name of the fluorescent dye [2-NBDG]. Which statistical test was applied to assess the results? Where is the scale bar? The picture in the Figure 7C [top left corner] resembles very much the picture in the Figure 1B in the Reference 40. Can you provide an original material or mention the source?
- Thank you for this comment. I have description in Figure 7 as you recommended.

Round 2
Reviewer 1 Report
The authors have addressed most of the points and the additional experiments have improved the quality of their work. However, it is difficult to support the antidiabetic effect of components of CLFs without measuring insulin signaling. I guess that these experiments are really important. I suggest eliminating or, at least, indicating the antidiabetic effects of components in the title and in the text as hypothesis.
Author Response
Dear. Sir (Reviewer 1),
Thank you for your generous decision. We have prepared a best-effort response to your comments, so we hope this is an appropriate response.
#Comments and Suggestions for Authors
The authors have addressed most of the points and the additional experiments have improved the quality of their work. However, it is difficult to support the antidiabetic effect of components of CLFs without measuring insulin signaling. I guess that these experiments are really important. I suggest eliminating or, at least, indicating the antidiabetic effects of components in the title and in the text as hypothesis.
- Thanks for your comments. As your pointed out, we have revised the manuscript title to “Aurones and flavonols from Coreopsis lanceolata L. flowers and their anti-oxidant, pro-inflammatory inhibition effects, and recovery effects on alloxan-induced pancreatic islets in zebrafish”. And in the text, it was replaced with the expression “potential anti-diabetic effect (or agents)” instead of "antidiabetic effect". We sincerely hope that this changes was appropriate to your comments.
I hope the improve version will be acceptable for publication in Molecules.
Yours sincerely
Prof. Nam-In Baek
